# Study on Risk Assessment of Flash Floods in Hubei Province

**Yong Tu [1,\*], Yanwei Zhao [2], Rui Dong [3], Han Wang [1], Qiang Ma [1], Bingshun He [1] and Changjun Liu [1]**

1    China Institute of Water Resource and Hydropower Research, Beijing 100038, China
2    College of Water Resources, North China University of Water Resources and Electric Power, Zhengzhou 450046, China
3    Beijing Tianzhixiang Information Technology Co., Ltd., Beijing 100191, China
\*    Correspondence: tuyong@iwhr.com

**Abstract:** Flash floods are typically associated with short, high-intensity and extreme rain-storms, and they are characterized by short response time and severely impact and damage communities in different areas in China. In order to scientifically assess the risks of flash floods, this paper takes Hubei Province as an example to carry out risk assessment. Based on Pearson correlation coefficient and principal component analysis methods, 14 factors were selected from 98 factors to establish a risk assessment model. The confidence coefficient model and multi-factor superposition method were used to determine the weight of each risk factor, and a risk map of Hubei Province was finally constructed. The results show that medium-high risk areas in Huanggang account for 47.00%, and high-risk areas account for 8.70%, with both areas adding up to more than 50%, followed by more than 40% in Shiyan, E'zhou and Xianning, and more than 30% in Huangshi, Yichang, Xiangyang, Jingmen and Suizhou. The risk level distribution is highly consistent with the location and frequency of flash flood disasters, shows high reliability, and can provide data support for flash flood disaster prevention and control. This study used a quantitative method to determine the key factors affecting flash flood disasters and provides a reference and basis for flash flood risk assessment in other provinces in China.

**Keywords:** flash floods; risk assessment; certainty factor model; regional evaluation; Hubei Province





## 1. Introduction

Flash floods are increasingly recognized as one of the most destructive natural hazards worldwide that cause disruption to the environment and communities, and most of them occur in remote mountainous areas with inconvenient transportation and poor communication, making it very difficult to issue forecasts and warnings [1]. Building resilience to flash floods requires knowledge of the socioeconomic characteristics of the local communities and their vulnerability to these extreme events [2]. Risk assessment is one of the essential components of risk management [3], and similar research studies have been carried out in such regions as Latin America, Australia, the Global South, and Central America [4–7].

Different approaches and methods have been established for risk assessment. The concepts of hazard, vulnerability and risks have been extensively used in various disciplines [8]. A new dynamic risk assessment method for flash flood disasters was proposed, combining the vulnerability of elements at risk with the same weight of comprehensive strength, frequency and loss [9]. Flood risk assessment consists of four essential elements, i.e., characterizing the areas, assessing hazards, assessing vulnerability and assessing risks [10]. According to the International Centre for Integrated Mountain Development (ICIMOD), flash flood hazard assessment includes two essential parts, i.e., assigning the flash flood intensity and the probability level of the hazard scenario. The Source-Pathways-Receptor-Consequence model of assessing risks was proposed by Gouldby and Samuals [11]. The first two components of risk (source and pathways) relate to hazard, and the last two (receptor and consequences) to vulnerability. Vulnerability describes the great possibility

of a receptor to suffer damage from a flash flood. Risk assessment is essential in making decisions about managing flash flood risks and can be carried out in four steps, i.e., characterizing the areas, assessing hazards, assessing vulnerability and assessing risks [12].

There are some other approaches to risk assessment. A socioeconomic vulnerability index was developed at the county level across the contiguous United States (CONUS) [2]. Integrated rainfall–runoff modeling (HEC-HMS) and hydraulic modeling (FLO-2D) schemes were used to assess flash flood inundation areas and depths under different rainfall scenarios in a mountainous watershed [13]. A unified system combining simulations of two impact forecasting methods, the Rapid Risk Assessment of the European Flood Awareness System (representing fluvial floods) and the radar-based ReAFFIRM method (representing flash floods), was explored [14].

Previous studies on flash-flood risk assessment had heightened demand for the advancement of more accurate models. The current research proposes state-of-the-art ensemble models of the boosted generalized linear model (GLMBoost) and random forest (RF) and the Bayesian generalized linear model (BayesGLM) methods for higher performance modeling [15]. Three machine learning models—genetic algorithm rule-set production, maximum entropy (MaxEnt), and random forest (RF)—were used for urban flood hazard maps with limited data [16]. The maximum entropy and frequency ratio methods, as well as the analysis of relationships between the flood events, were used for flood risk assessment in Iran [17]. A robust method coupling the maximum entropy (MAXENT) and the future land use simulation (FLUS) model was used for predicting future waterlogging-prone areas [18].

From the above studies, it can be seen that research on the risk assessment of flood disasters has mainly focused on risk assessment models, risk indicators and risk indices for flash floods. Due to the lack of data, most of the articles are based on the risk analysis of small watersheds or qualitative research using limited indicators, and there are few discussions of quantitative research on the risk assessment of flash flood disasters over a large administrative region. China launched the Flash Flood Disasters Investigation and Evaluation project from 2012 to 2016, covering 30 provinces, 305 cities and 2138 counties, involving a total land area of 7.55 million km$^2$ and a population of nearly 900 million [19,20]. Through general census and comprehensive evaluations, a rich dataset on human settlements, underlying surface conditions, and the social and economic situations of flash flood disasters was collected, making it easier for us to describe the risks of flood disasters in more detail and create a risk assessment model for subsequent flood disasters.

The purpose of this study was to find a feasible risk assessment method based on the Flash Flood Disasters Investigation and Evaluation dataset, reveal the causes of flash flood disaster formation to a certain extent, and identify the key factors affecting flood disasters by making full use of the existing data. The Pearson correlation coefficient mainly analyzes the linear correlation between several factors [21]. Principal component analysis is to describe the most important characteristics of the data with the least data. These calculations are to reduce the data dimensions and eliminate the collinearity impact of the data. Through a series of dimensionality reduction processing, it was helpful for us to find the rules of flash flood formation from the basic data. The CF-based multi-factor superposition method adopted in this manuscript was initially applied to the risk assessment of geological disasters in China [22]. Its research areas include plateaus, mountains, hills and watersheds and involve a variety of landforms and complex geological structures that bear many similarities with areas of flash floods in China.

Hubei Province is one of the areas prone to flash flood disasters in China. Based on the Flash Flood Disasters Investigation and Evaluation data for Hubei Province, this paper constructed a flash flood disaster risk assessment model and drew a flash flood disaster risk map to provide a reference and basis for flash flood risk assessment in other provinces in China.

## 2. Materials and Methods

### 2.1. Study Area

Hubei Province is located in the middle reaches of the Yangtze River and north of Dongting Lake. It extends about 740 km from east to west and 470 km from north to south, and covers an area of 185,900 km$^2$, accounting for 1.94% of the total area of China. It is located in the transition zone from the second step to the third step in China's terrain. The entire landform of Hubei is an incomplete watershed with three sides high, the middle low, open to the south, and a gap in the north. The province has various types of landforms, including mountains, hills and plains. Among the landform types in Hubei, the mountainous area is the most extensive, covering about 104,000 km$^2$ and accounting for 55.5% of the total area of the province; the hills cover about 45,000 km$^2$, accounting for 24.5%; and the plains cover about 40,000 km$^2$, accounting for 20%. Figure 1 shows the location of Hubei Province.

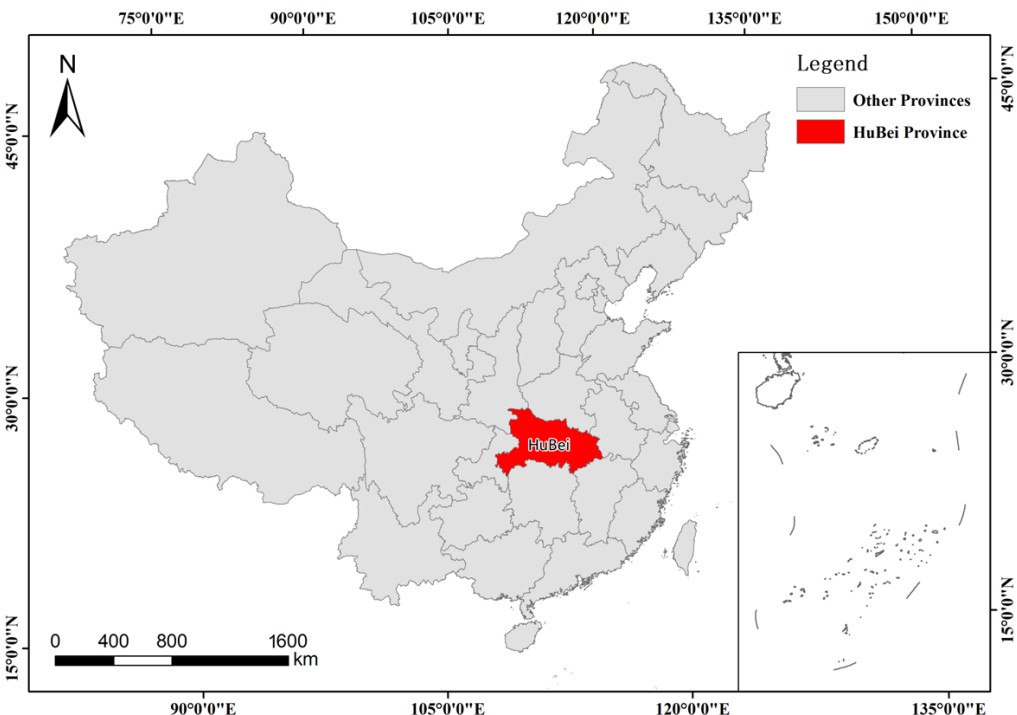

**Figure 1.** Location of Hubei Province.

According to statistics, since the founding of New China, a total of 1544 flash flood disasters have occurred in 75 flash flood disaster prevention counties in the province, causing 6763 deaths, 448 missing, 1.91 million houses damaged, and direct economic losses of RMB 39.5 billion. According to the Flash Flood Disasters Investigation and Evaluation results, the area of flash flood disaster prevention and control in Hubei Province is 126,700 km$^2$, involving 13 cities, 75 counties, and 795 townships in the whole province. Figure 2 shows the mountainous and riverside villages most vulnerable to flash floods. Some of the villages have suffered one or more flash floods during the period from 1949 to 2015. These villages are the key targets for the prevention of flash floods, and the investigation and evaluation work were carried out from 2013 to 2015.

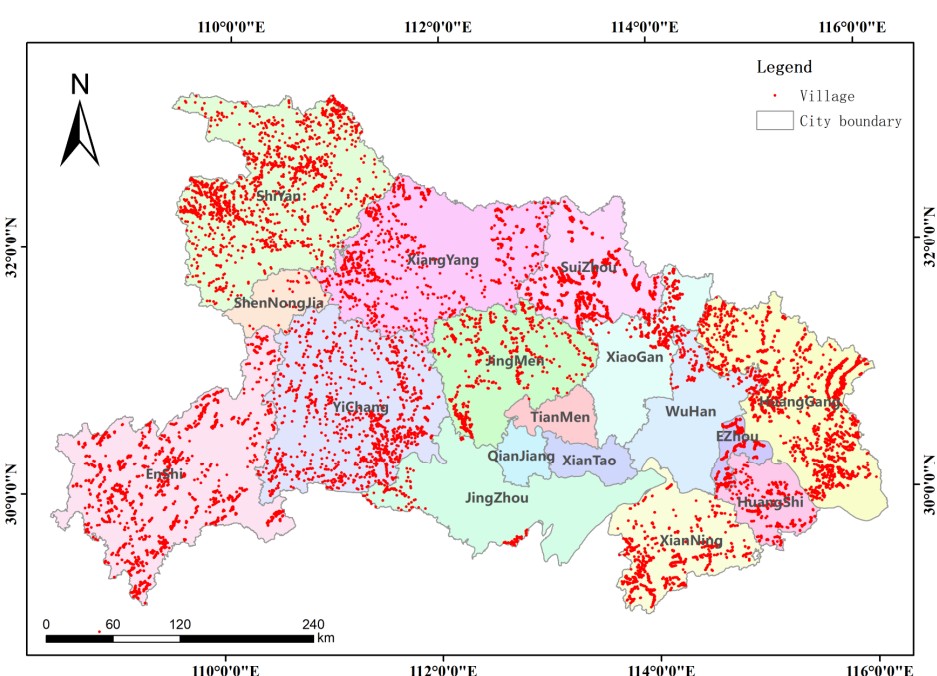

**Figure 2.** The Villages Affected by Flash Floods in Hubei Province.

*2.2. Data Collection*

The Flash Flood Disasters Investigation and Evaluation project has been the largest non-engineering water conservancy project in China since the founding of new China. It is also the largest-scale general census on disaster background in the flood management and mitigation fields. The whole project lasted 4 years, from 2012 to 2016, covering 30 provinces, 305 cities and 2138 counties and involving a total land area of 7.55 million km$^2$ and a population of nearly 900 million.

Extensive flood investigations and risk assessments were carried out in Hubei Province, and the investigation and evaluation work was mainly carried out and completed in 2013–2015. The investigation and evaluation results for flood disasters in Hubei Province were supplemented and sorted out, and 88 items in seven categories of relatively comprehensive flood disaster investigation and evaluation results were formed. In 2013–2015, a total of 74 counties (cities and districts) in Hubei Province were investigated and evaluated, involving a total population of 42.114 million and an area of 160,000 km$^2$. In the control area, the total population was 16.34 million, and the total land area was 126,700 km$^2$. The investigation covered 44,000 flash flood hazard areas in the province, with a total population of 3.92 million, involving 7,189 enterprises and institutions and 1.12 million households in hazard areas. On this basis, social and economic surveys were conducted on 44,608 villages in the control area, and the regional distributions of 1544 historical floods, 3240 automatic monitoring stations, 8868 wireless early warning transmitting stations, 794 simple hydrological gauging stations, and 12,012 simple rainfall gauging stations were analyzed. Moreover, detailed surveys and investigations were carried out on the riverside villages, and 11,818 longitudinal sections and 35,318 cross sections of ditches where riverine villages are located were measured and collated, providing valuable basic data for the prevention and control of flash flood disasters in Hubei Province.

The basic dataset and result dataset of the Flash Flood Disasters Investigation and Evaluation in Hubei Province were collected. In the basic dataset, Hubei Province was divided into 12,321 small watersheds (Figure 3). We could collect the area, slope, elevation, catchment time, flood peak modulus and other attributes of each watershed from the layer. The basic dataset contained land use data and soil type data with a scale of 1:50,000 that can be used to analyze the land use and soil type information of each watershed. The result dataset contained all of the investigation and evaluation results, including flood

survey, river channel survey, population and property distribution, flood control capacity evaluation, etc. According to statistics, from 1949 to 2015, a total of 11,925 flash floods occurred in 3649 villages in Hubei Province, and a heat map of flash flood disasters was drawn accordingly (Figure 4). The red area represents a high frequency of flash flood disasters, and the blue area represents a low frequency of flash flood disasters.

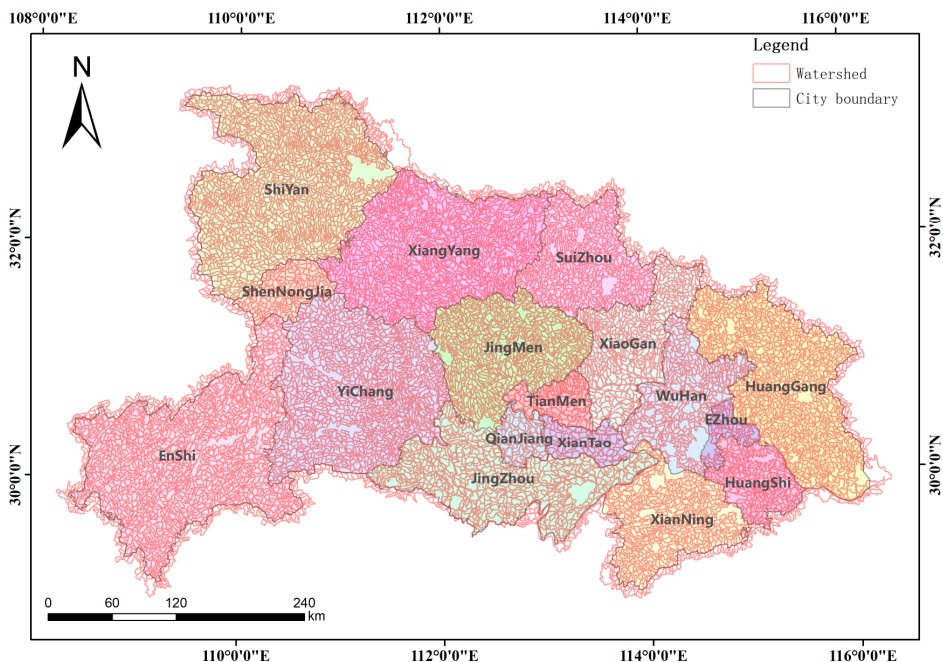

**Figure 3.** Map of Watershed Division.

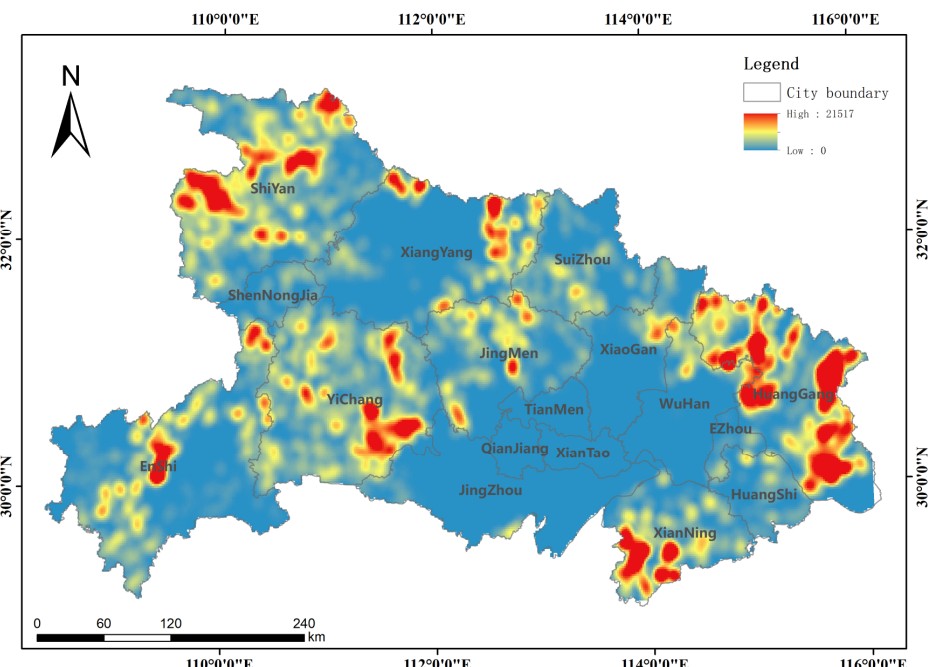

**Figure 4.** Heat Map of Flash Flood Disasters.

In order to study the characteristics of rainstorms in Hubei Province, we compiled a contour map of mean annual rainfall and variation coefficients for 1, 6 and 24 h in Hubei Province. At the same time, a rainstorm atlas and distribution map of hydro-meteorological regions and stations were compiled (Figure 5).

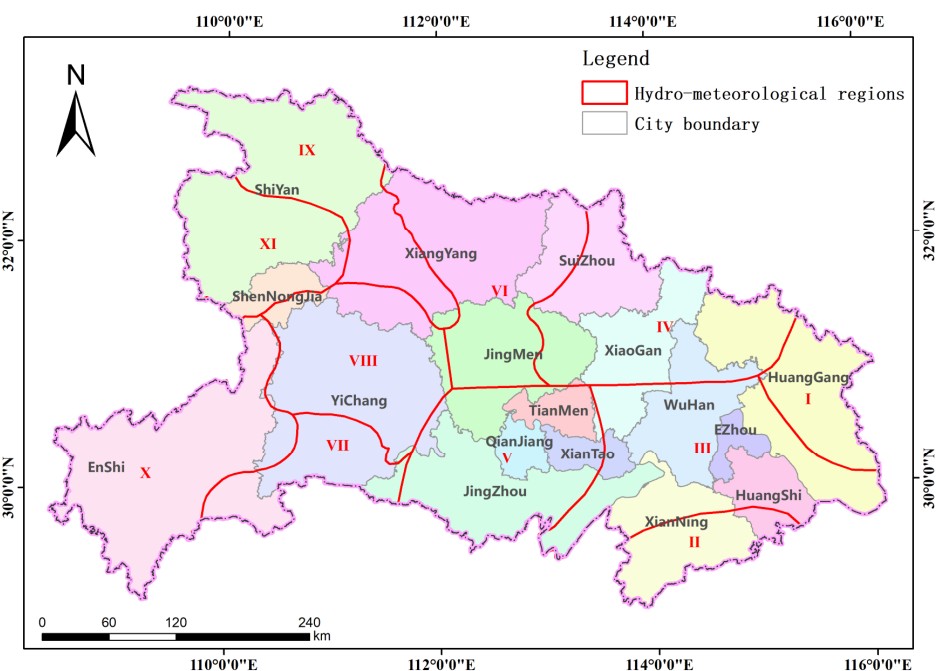

**Figure 5.** The Hydro-meteorological Regions in Hubei Province.

### *2.3. Modelling Approaches*

### 2.3.1. Risk Conceptual Model

According to the theory of natural disaster systems, nature (N) and society (S) are the essential characteristics of the flash flood disaster system [23], so a flash flood disaster (FFD) can be defined as FFD = N∩S.

It can be concluded that the flash flood disaster system is an environmental change system composed of disaster-causing factors, disaster-causing environment and disaster-affected bodies. Flash flood hazards occur when the three components of the system interact. Therefore, the flash flood disaster system consists of three parts: disaster-causing factors, disaster-causing environment and disaster-affected subjects.

$$FFDS = FFC∩FFCE∩FFAS \qquad (1)$$

FFC represents the disaster-causing factors, which are the initial condition of a flood disaster; FFAS refers to the disaster-affected subjects, which are the object factors of a flash flood disaster; FFCE is the disaster-causing environment, which is relatively stable in a certain period of time and can be used as the breeding environment of a flood disaster. The three elements of disaster-causing factors, disaster-causing environment, and disaster-affected bodies that constitute a flash flood disaster system can be specifically expressed as precipitation factors, underlying surface conditions of small watersheds, and socioeconomic conditions. Therefore, the flash flood disaster system is a three-dimensional small watershed spatial variation system composed of precipitation factors, small watershed underlying surface area factors, and socioeconomic factors. When these three factors combine, flash flood disasters occur. The precipitation factor and the underlying surface area factor of the small watersheds reflect the natural characteristics of the flash flood disaster system. The socioeconomic factors characterize the social attributes of the flash flood disaster system.

$$FFSM = FFR∩FFUS∩FFSE \qquad (2)$$

The risk of flash flood disasters refers to the scenario of loss and damage to human life and property caused by a flash flood due to heavy rain in hilly areas in the future. Since it is difficult to describe the scenarios of future adverse events with precise mathematical

language, the concept of flood disaster risks was introduced to express the risk of a flood disaster quantitatively (Figure 6).

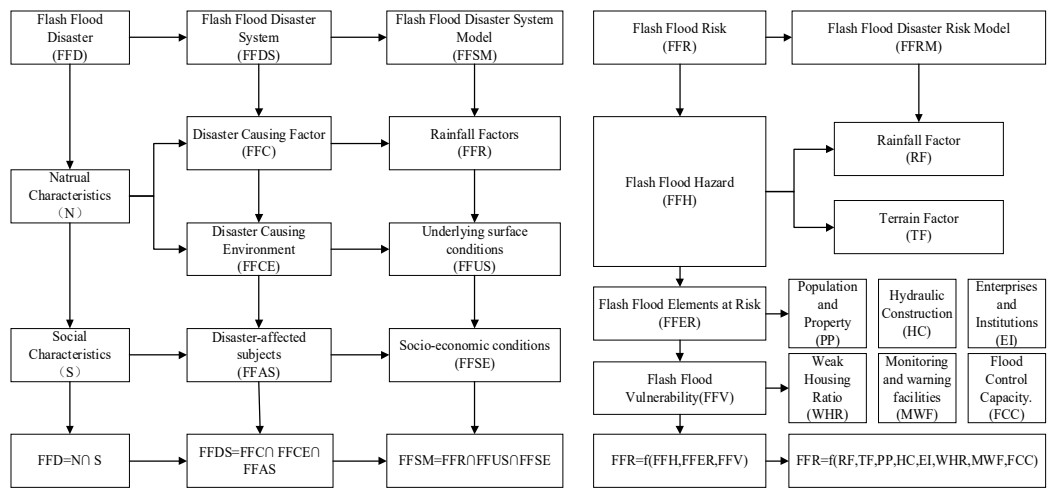

**Figure 6.** Conceptual Model of the Flash Flood Risk System.

The risk of flash flood (FFR) is a function of flash flood hazard (FFH), flash flood elements at risk (FFER), and flash flood vulnerability (FFV). The hazard of flash flood disasters is mainly reflected in two aspects: rainfall factors (RF) and terrain factors (TF). The elements at risk in flash flood disasters are mainly risk-bearing and hazard-bearing objects, and the most affected factors are population and property (PP), hydraulic construction (HC) and enterprises and institutions (EI), which are closely related to the lives and properties of the people. Flash flood vulnerability (FFV) refers to the ability of society and the environment to resist flash flood disasters, which can be measured by factors such as weak housing ratio (WHR) monitoring and warning facilities (MWF) and flood control capacity (FFC).

So finally, the flash flood risk can be expressed as a function of the following factors [23].

$$FFR = f (RF, TF, PP, HC, EI, WHR, MWF, FCC) \tag{3}$$

2.3.2. Risk Computational Model

Based on the above basic understanding of the risks of flash flood disasters, it is necessary to use the data from Flash Flood Disasters Investigation and Evaluation results to further identify the risk factors for flash flood disasters and determine the risks, elements at risk and vulnerabilities in the flash floods, including flood disasters, quantitative expression of flood disaster risks, derivation and prediction of key factors of disaster risks, and the degree of possible risks.

The data from Flash Flood Disasters Investigation and Evaluation results mainly include information such as rainfall, subsoil, socioeconomic and historical flash flood disasters, and a database for identifying risk factors for flash flood disasters.

Considering the large amount of data in the Flash Flood Disasters Investigation and Evaluation results, including a large number of data elements, for the subsequent identification and extraction of flood disaster risk factors, it is necessary to use an empirical knowledge discrimination method to initially verify the existing data and identify factors strongly associated with flood disaster risks. The data not closely related to the disaster location and vulnerability were preliminarily screened out according to the risk components of flash flood disasters (Figure 7).

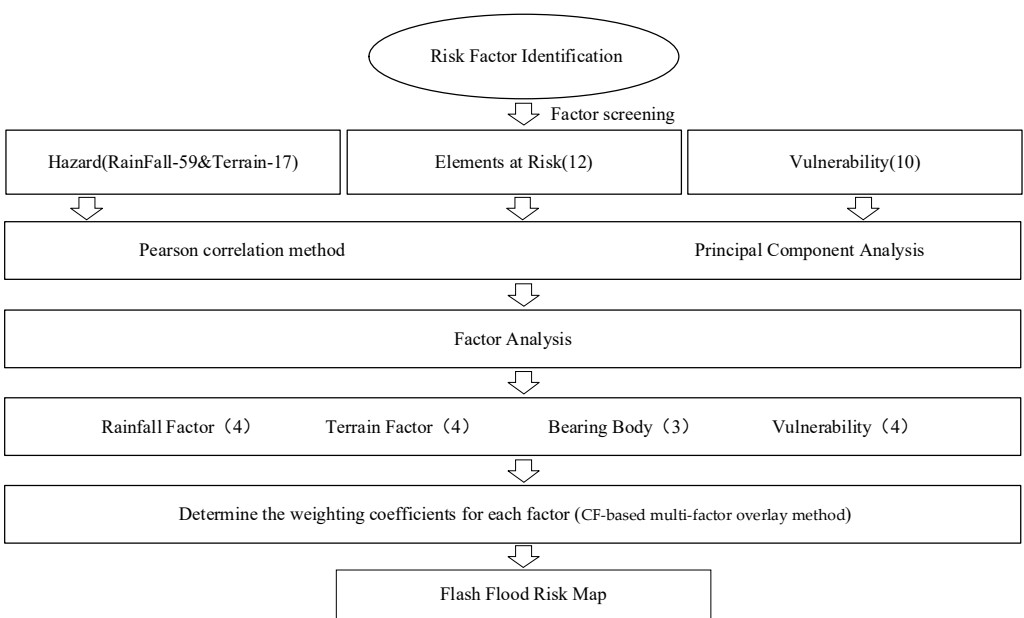

**Figure 7.** Flowchart of the Risk Computational Model.

(1) Pearson correlation coefficient

The Pearson correlation coefficient $r_{xy}$ between two n-dimensional variables x and y is defined as the quotient of the covariance and standard deviation between the two variables [21].

$$r_{xy} = \frac{\sum_{i=1}^{n} x_i y_i - n\overline{xy}}{(n-1)s_x s_y} = \frac{n\sum_{i=1}^{n} x_i y_i - \sum_{i=1}^{n} x_i \sum_{i=1}^{n} y_i}{\sqrt{n\sum_{i=1}^{n} x_i^2 - \left(\sum_{i=1}^{n} x_i\right)^2} \sqrt{n\sum_{i=1}^{n} y_i^2 - \left(\sum_{i=1}^{n} y_i\right)^2}} \tag{4}$$

The value range of the Pearson correlation coefficient is −1 to 1. A value closer to 1 indicates a positive correlation, a value closer to −1 indicates a negative correlation, and a value closer to 0 indicates insignificant correlation. The Pearson correlation coefficient method is mainly used to eliminate the influence of multi-collinearity.

(2) Principal component analysis

Principal component analysis (PCA) is an unsupervised learning problem. This is a common resource reduction method. Dimensionality reduction is performed by mapping high-dimensional data onto the axis with the highest variance and discarding the axis with the lowest variance [24]. The main idea is to reduce a set of n-dimensional vector data to k-dimensional uncorrelated variables, namely principal components. The specific goal is to transform the raw data into an orthonormal basis such that the pairwise covariance of each domain is 0, the variance of the domains is as large as possible, and the projections of the raw data can be distributed in independent directions. Since it is as dispersed as possible, more original information can be preserved, and the search directions are independent to avoid redundancy of preserved information.

(3) Factor analysis

Factor analysis is a statistical technique for extracting common factors from groups of variables. It can simplify and reduce the dimensionality of multidimensional variables, self-optimize and find hidden representative factors among many variables, classify common variables into one factor, and reflect most of the information of the original data with fewer

factors [25]. The mean of n-dimensional population $X = \{x_1, x_2, \ldots, xn\}$ is $u = (u_1, u_2, \ldots, u_n)$. The established factor analysis mathematical model is as follows:

$$x_1 = u_1 + a_{11}f_1 + a_{12}f_{2+\ldots}+ a_{1m}f_{m+\ldots+\varepsilon_1}$$
$$x_2 = u_2 + a_{21}f_1 + a_{22}f_{2+\ldots}+ a_{2m}f_{m+\ldots+\varepsilon_2}$$
$$\ldots$$
$$x_n = u_n + a_{n1}f_1 + a_{n2}f_{2+\ldots}+ a_{nm}f_{m+\ldots+\varepsilon_n}$$

(5)

where $f_j$ ($j = 1, 2, \ldots, m$) is common factors; $\varepsilon_i$ is a special factor unique to variable $x_i$ ($i = 1, 2, \ldots, n$), and they are unobservable hidden variables. $a_{ij}$ ($i = 1, 2, \ldots, n; j = 1, 2, \ldots, m$) is the load on the common factor $f_j$ of variable $x_i$, which reflects the importance of the common factor to the variable and plays an important role in explaining the common factor.

(4)    Certainty factor model

The certainty factor model (CF) was used to analyze the weight of each factor. This model was originally developed by Shortliffe et al. [26] and improved by Heckerman [27]. The formula is as follows:

$$CF = \begin{cases} \frac{pp_a - pp_s}{pp_a(1 - pp_s)}, & pp_a - pp_s \\[2mm] \frac{pp_a - pp_s}{pp_s(1 - pp_a)}, & pp_a - pp_s \end{cases}$$

(6)

where $PPa$ is the conditional probability of occurrence of events (flash flood disasters) in Type-a data and is the ratio of the number of flash flood disasters in Type-a data to the area of Type-a; $PPs$ is the prior probability of flash flood disasters in the entire study area, which can be expressed as the ratio of the number of flash flood disasters in the entire study area to the area of the study area.

The certainty factors ($CF$) of each risk layer are superimposed in pairs and combined separately; if the certainty factors of the two factor layers are $x$ and $y$, respectively, and the combined result is $Z$, then the combined formula is expressed as follows:

$$Z = \begin{cases} x + y - xy, & x \geq 0 \ and \ y \geq 0 \\[3mm] \frac{x+y}{1 - \min(|x|, |y|)}, & xy < 0 \\[3mm] x + y + xy, & x < 0 \ and \ y < 0 \end{cases}$$

(7)

In order to facilitate interpretation of the comprehensive results, the comprehensive $CF$ value can be divided into five levels (Table 1), which represent the smallest, small, uncertain, large, and largest contribution to the occurrence of flash floods.

**Table 1.** Division of CF Value.

| Order Number | The Range of CF Value | Contribution to Disasters | Consider All Factors | Consider All Factors Except Factor A |
|---|---|---|---|---|
| 1 | $<-0.6$ | smallest | $Z_{all-1}$ | $Z_{all-A1}$ |
| 2 | $-0.6 \sim -0.2$ | small | $Z_{all-2}$ | $Z_{all-A2}$ |
| 3 | $-0.2 \sim 0.2$ | uncertain | $Z_{all-3}$ | $Z_{all-A3}$ |
| 4 | $0.2 \sim 0.6$ | large | $Z_{all-4}$ | $Z_{all-A4}$ |
| 5 | $\geq 0.6$ | largest | $Z_{all-5}$ | $Z_{all-A5}$ |

A CF-based multi-factor superposition method was proposed to determine the factor weights of hazards [22]. The method can be divided into three steps: Each factor to hazards is quantified with $CF$, so that the value of each factor is in the range between 0~1. All factors are progressively superimposed to calculate the relative contribution of all factors $Z_{all}$. For

each calculation factor A, all other factors are progressively superimposed to obtain the relative contribution of other factors except it ($Z_{all-A}$). By the subtraction method, we can obtain the relative contribution value of the calculation factor. The relative contributions of each factor in different value ranges are accumulated to obtain the total contribution ($T_A$) of each factor, and the contribution of each factor is normalized to obtain the weight of each factor.

$$T_A = \sum_{i=1}^{5} |Z_{all-i} - Z_{all-Ai}| \tag{8}$$

## 3. Risk Analysis

### 3.1. Risk Factor Identification

3.1.1. Hazard

Since local short-term heavy rainfall is the main factor causing flash flood disasters, the consideration of precipitation factors was mainly based on the design rainstorm results for small watersheds, and the statistical design rainstorm parameters of typical periods and typical frequencies were selected as candidate precipitation factors. Because the runoff process of flood precipitation in a small catchment area is very short, generally no more than 24 h, the typical precipitation duration is 10 min, 60 min, 3 h, 6 h and 24 h. The planned rainstorm frequency is usually once in 5 years, once in 10 years, once in 20 years, once in 50 years and once in 100 years. The statistical parameters of the design rainstorm are mainly the mean value of each typical time period, design rainstorm with typical frequency, variation coefficient and modulus coefficient. A total of 59 factors were used as candidate precipitation factors.

The terrain factors of flash flood disaster risks mainly involve factors related to small watersheds and underlying surface conditions. The division of small watersheds and the extraction of basic attributes are the premise and foundation. In this study, a total of 17 parameters directly related to the basic attributes of the watersheds and the attributes of flow were selected as candidate terrain factors (Table 2).

**Table 2.** Attributes of Rainfall and Terrain in Small Watersheds.

| Rainfall Factors | Description | Terrain Factors | Description |
|---|---|---|---|
| $H_{10min}$ | Average annual maximum 10 min rainfall | WSAREA | Area of small watershed |
| $Cv_{10min}$ | Variation coefficient of 10 min rainfall | WSPERI | Perimeter of small watershed |
| $H_{10min\_20\%}$ | 5-return-year design storm rainfall of 10 min | WSSLP | Slop of small watershed |
| $Kp_{10min\_20\%}$ | Modulus coefficient of 10 min and 5-return-year design rainstorm | WSSHPC | Shape coefficient of small watershed |
| $H_{10min\_10\%}$ | 10-return-year design storm rainfall of 10 min | MAXLEN | The longest catchment path length of small watershed |
| $Kp_{10min\_10\%}$ | Modulus coefficient of 10 min and 10-return-year design rainstorm | MAXLSLP | Gradient of the longest catchment path length in small watershed |
| $H_{10min\_5\%}$ | 20-return-year design storm rainfall of 10 min | MAXLS1085 | 10–85% gradient of the longest catchment path length in small watershed |
| $Kp_{10min\_5\%}$ | Modulus coefficient of 10 min and 20-return-year design rainstorm | CENTERELV | Centroid elevation of small watershed |
| $H_{10min\_2\%}$ | 50-return-year design storm rainfall of 10 min | OUTLETELV | Elevation of small watershed outlet |
| $Kp_{10min\_2\%}$ | Modulus coefficient of 10 min and 50-return-year design rainstorm | OUTLETAD8 | Total catchment area of small watershed outlet |
| $H_{10min\_1\%}$ | 100-return-year design storm rainfall of 10 min | MAXELV | Highest elevation of small watershed |
| $Kp_{10min\_1\%}$ | Modulus coefficient of 10 min and 100-return-year design rainstorm | RVLEN | Length of reach in small watershed |
| $Cv_{60min}$ | Variation coefficient of 1 h rainfall | RVSLP | Reach gradient of small watershed |
| $H_{60min\_20\%}$ | 5-return-year design storm rainfall of 1 h | MFP | Flood peak modulus of small watershed |
| $Kp_{60min\_20\%}$ | Modulus coefficient of 1 h and 5-return-year design rainstorm | NSTEPS | Concentration time of small watershed |
| $H_{60min\_10\%}$ | 10-return-year design storm rainfall of 1 h | AVEROU | Average roughness of slope surface |
| $Kp_{60min\_10\%}$ | Modulus coefficient of 1 h and 10-return-year design rainstorm | AVRINF | Average steady seepage rate. |

### 3.1.2. Elements at Risk

Flash flood hazard elements at risk are those who bear the risks of flash flood hazards and who may be affected by physical, economic and social hazards (Table 3), including flash flood disaster prevention and control areas, wading projects, enterprises and institutions, etc.

**Table 3.** Statistics on Households, Enterprises and Projects in Small Watersheds.

| Elements at Risk | Description | Elements at Risk | Description |
|---|---|---|---|
| PC_A | Population | RES | Amount of reservoir |
| HTC_A | Housing | SLUICE | Amount of sluice |
| ETC_A | Family property | BRIDGE | Amount of bridge |
| BUSINESS | Amount of business | DAM | Amount of dam |
| SCHOOL | Amount of school | CULVERT | Amount of culvert |
| HOSPITAL | Amount of hospital | NURSINGHOME | Amount of nursing home |

### 3.1.3. Vulnerability

The vulnerability factors of flood disasters mainly include disaster prevention factors and vulnerability factors. Disaster prevention factors mainly include current flood control capacity, population affected by floods of different frequencies, monitoring and early warning facilities and equipment, dam construction, quantity of flood control materials, backup communication equipment and communication methods, resettlement sites, resettlement capabilities, grassroots organization systems, public defense awareness, etc. There is a negative correlation between disaster prevention capacity and vulnerability. The stronger the disaster prevention capability, the weaker the vulnerability of the disaster system. Vulnerability factors mainly include the proportion of weak buildings among residential buildings, which is positively correlated with vulnerability (Table 4).

**Table 4.** Statistics on Flood Prevention Capacity in Small Watershed.

| Vulnerability Factors | Description | Vulnerability Factors | Description |
|---|---|---|---|
| RTC3_A | Proportion of III-type houses with poor quality | FHNL | Flood control capacity |
| RTC4_A | Proportion of IV-type houses with poor quality | DIKE | Flood control capacity of embankment |
| ZD_WATA | Density of automatic rainfall gauging stations | PC_A5 | Population affected by 5-return-year flood |
| JY_YL | Density of simple rainfall monitoring sites | PC_A20 | Population affected by 20-return-year flood |
| JY_SW | Density of simple hydrological monitoring sites | PC_A100 | Population affected by 100-return-year flood |

### 3.2. Risk Factor Screening

Based on the Flash Flood Disasters Investigation and Evaluation results, and using the empirical discrimination method, 76 flood disaster hazard factors, including 59 precipitation factors and 17 terrain factors; 12 elements at risk; and 10 vulnerability factors were preliminarily screened out, adding up to 98 factors. Since too many factors lead to over-interpretation of flash flood risks, it is necessary to reduce the dimensionality of factors and identify the representative factors that can best explain flash flood risks.

#### 3.2.1. Pearson Correlation Coefficient

For risk factors such as the rainfall factor, terrain factor, elements at risk, and vulnerability factor, the Pearson correlation coefficient among the factors was calculated. It can be seen from Figure 8a that the correlation between rainfall factors was high, and most of them were strongly positively correlated. Similarly, there was a strong positive correlation between the variation coefficient and modulus coefficient. Correspondingly, there was a negative correlation between these two types of data. It can be seen from Figure 8b that the elevation factor and slope factor were strongly positively correlated, and they were negatively correlated with the average roughness. From Figure 8c,d, the correlation

between the elements at risk was not obvious, which was also true for the relationship between vulnerability factors.

(**a**)Rainfall factor

(**b**)Terrain factor

(**c**) Elements at risk

(**d**) Vulnerability factor

**Figure 8.** Pearson Correlation Coefficient Matrix.

3.2.2. Principal Component Analysis

PCA attempts to recombine the original variables into a new set of variables consisting of several uncorrelated generic variables. At the same time, if necessary, some variables with low total counts can be removed to represent information about the original variable with as much knowledge as possible. Principal component analysis is also a way to deal with dimensionality reduction.

Table 5 shows the cumulative PCA contributions of precipitation, terrain, elements at risk and vulnerability factors. It can be seen from the table that the cumulative contribution of 3–4 principal components of each type of factor can reach 70~90%, which means that 3–4 types of important dimensional data can describe the main characteristics of the dataset.

**Table 5.** Cumulative Contribution of Principal Components.

| Principal Component | Cumulative Contribution (%) | | | |
|---|---|---|---|---|
| | Rainfall | Terrain | Elements at Risk | Vulnerability |
| Com1 | 41.65 | 32.08 | 26.19 | 26.93 |
| Com2 | 72.07 | 50.49 | 47.30 | 43.82 |
| Com3 | 84.93 | 66.28 | 61.86 | 63.62 |
| Com4 | 92.05 | 72.79 | 71.20 | 73.13 |
| Com5 | 95.62 | 78.34 | 76.40 | 79.71 |
| Com6 | 98.03 | 83.38 | 83.92 | 85.74 |
| Com7 | 99.41 | 87.58 | 89.44 | 90.89 |
| Com8 | 99.84 | 91.40 | 94.37 | 94.00 |
| Com9 | 99.93 | 94.28 | 98.43 | 97.62 |
| Com10 | 99.96 | 96.76 | 99.46 | 100.00 |

### 3.2.3. Factor Analysis

Principal component analysis uses a new low-dimensional vector instead of the original data for analysis, which does not help reveal the intrinsic relationship between each attribute and flash flood risk. Factor analysis can analyze the contribution of each element to the whole. We used factor analysis to select elements contributing more to the principal components of the risk calculation. Tables 6 and 7 are the factor analysis results for precipitation and terrain elements in the domain.

**Table 6.** Analysis Results for Rainfall Factors.

| Rainfall | Com 1 | Com 2 | Com 3 | Com 4 | Rainfall | Com 1 | Com 2 | Com 3 | Com 4 |
|---|---|---|---|---|---|---|---|---|---|
| $H_{10min}$ | 0.496 | −0.175 | −0.092 | 0.813 | $CV_{10min}$ | −0.605 | 0.498 | 0.165 | 0.102 |
| $H_{10min\_20\%}$ | 0.449 | −0.128 | −0.077 | 0.863 | $CV_{1h}$ | −0.272 | 0.318 | 0.774 | −0.089 |
| $H_{10min\_10\%}$ | 0.384 | −0.064 | −0.056 | 0.914 | $CV_{6h}$ | 0.026 | 0.941 | 0.337 | −0.002 |
| $H_{10min\_5\%}$ | 0.324 | −0.009 | −0.038 | 0.944 | $CV_{24h}$ | −0.063 | 0.713 | −0.121 | −0.048 |
| $H_{10min\_2\%}$ | 0.253 | 0.051 | −0.017 | 0.964 | $KP_{10min\_20\%}$ | −0.610 | 0.491 | 0.165 | 0.105 |
| $H_{10min\_1\%}$ | 0.207 | 0.089 | −0.004 | 0.968 | $P_{10min\_10\%}$ | −0.606 | 0.496 | 0.165 | 0.102 |
| $H_{1h}$ | 0.496 | −0.175 | −0.092 | 0.813 | $P_{10min\_5\%}$ | −0.604 | 0.498 | 0.165 | 0.101 |
| $H_{1h\_20\%}$ | 0.449 | −0.128 | −0.077 | 0.863 | $KP_{10min\_2\%}$ | −0.603 | 0.499 | 0.165 | 0.101 |
| $H_{1h\_10\%}$ | 0.384 | −0.064 | −0.056 | 0.914 | $P_{10min\_1\%}$ | −0.602 | 0.500 | 0.165 | 0.101 |
| $H_{1h\_5\%}$ | 0.324 | −0.009 | −0.038 | 0.944 | $KP_{1h\_20\%}$ | −0.269 | 0.313 | 0.772 | −0.104 |
| $H_{1h\_2\%}$ | 0.253 | 0.051 | −0.017 | 0.964 | $KP_{1h\_10\%}$ | −0.272 | 0.317 | 0.773 | −0.092 |
| $H_{1h\_1\%}$ | 0.207 | 0.089 | −0.004 | 0.968 | $KP_{1h\_5\%}$ | −0.272 | 0.318 | 0.774 | −0.089 |
| $H_{3h}$ | 0.773 | −0.149 | 0.356 | 0.457 | $KP_{1h\_2\%}$ | −0.272 | 0.319 | 0.776 | −0.086 |
| $H_{3h\_20\%}$ | 0.747 | −0.113 | 0.453 | 0.445 | $KP_{1h\_1\%}$ | −0.272 | 0.319 | 0.780 | −0.085 |
| $H_{3h\_10\%}$ | 0.694 | −0.059 | 0.574 | 0.422 | $KP_{3h\_20\%}$ | 0.035 | 0.894 | 0.410 | 0.023 |
| $H_{3h\_5\%}$ | 0.643 | −0.019 | 0.653 | 0.398 | $KP_{3h\_10\%}$ | 0.023 | 0.930 | 0.365 | 0.010 |
| $H_{3h\_2\%}$ | 0.586 | 0.020 | 0.718 | 0.371 | $KP_{3h\_5\%}$ | 0.019 | 0.934 | 0.355 | 0.008 |
| $H_{3h\_1\%}$ | 0.551 | 0.041 | 0.750 | 0.354 | $KP_{3h\_2\%}$ | 0.016 | 0.936 | 0.350 | 0.008 |
| $H_{6h}$ | 0.927 | −0.225 | −0.040 | 0.291 | $KP_{3h\_1\%}$ | 0.015 | 0.937 | 0.348 | 0.008 |
| $H_{6h\_20\%}$ | 0.945 | −0.128 | 0.009 | 0.296 | $KP_{6h\_20\%}$ | 0.068 | 0.926 | 0.341 | −0.021 |
| $H_{6h\_10\%}$ | 0.951 | 0.053 | 0.078 | 0.291 | $KP_{6h\_10\%}$ | 0.035 | 0.940 | 0.339 | −0.006 |
| $H_{6h\_5\%}$ | 0.932 | 0.185 | 0.128 | 0.282 | $KP_{6h\_5\%}$ | 0.027 | 0.941 | 0.337 | −0.002 |
| $H_{6h\_2\%}$ | 0.897 | 0.305 | 0.171 | 0.268 | $KP_{6h\_2\%}$ | 0.022 | 0.941 | 0.336 | 0.000 |
| $H_{6h\_1\%}$ | 0.870 | 0.368 | 0.194 | 0.258 | $KP_{6h\_1\%}$ | 0.019 | 0.941 | 0.336 | 0.001 |
| $H_{24h}$ | 0.927 | −0.239 | −0.186 | 0.211 | $KP_{24h\_20\%}$ | 0.019 | 0.690 | −0.141 | −0.058 |
| $H_{24h\_20\%}$ | 0.950 | −0.161 | −0.157 | 0.210 | $KP_{24h\_10\%}$ | −0.046 | 0.711 | −0.126 | −0.051 |
| $H_{24h\_10\%}$ | 0.971 | −0.013 | −0.101 | 0.210 | $KP_{24h\_5\%}$ | −0.061 | 0.713 | −0.122 | −0.049 |
| $H_{24h\_5\%}$ | 0.971 | 0.099 | −0.057 | 0.206 | $KP_{24h\_2\%}$ | −0.071 | 0.713 | −0.120 | −0.047 |
| $H_{24h\_2\%}$ | 0.956 | 0.208 | −0.013 | 0.200 | $KP_{24h\_1\%}$ | −0.075 | 0.713 | −0.118 | −0.047 |
| $H_{24h\_1\%}$ | 0.941 | 0.269 | 0.012 | 0.196 | | | | | |

**Table 7.** Results of Analysis of Terrain Factors.

| Terrain | Com 1 | Com 2 | Com 3 | Com 4 | Com 5 | Com 6 | Com 7 | Com 8 |
|---|---|---|---|---|---|---|---|---|
| WSAREA | 0.012 | 0.808 | −0.085 | 0.054 | −0.001 | 0.084 | 0.006 | −0.013 |
| WSPERI | −0.049 | 0.589 | 0.429 | −0.132 | 0.017 | 0.017 | 0.009 | 0.015 |
| WSSLP | 0.420 | 0.013 | −0.044 | 0.041 | 0.666 | −0.280 | 0.028 | 0.037 |
| WSSHPC | 0.103 | 0.927 | −0.236 | 0.112 | 0.066 | −0.023 | 0.954 | 0.000 |
| MAXLEN | −0.076 | 0.721 | 0.604 | −0.239 | 0.039 | −0.038 | −0.199 | 0.020 |
| MAXLSLP | 0.844 | −0.035 | −0.099 | 0.036 | 0.389 | −0.053 | 0.076 | 0.233 |
| MAXLS1085 | 0.916 | −0.038 | −0.088 | 0.024 | 0.347 | −0.021 | 0.074 | 0.097 |
| CENTERELV | 0.268 | 0.012 | 0.008 | 0.003 | 0.907 | −0.125 | 0.010 | 0.225 |
| OUTLETELV | 0.132 | −0.007 | −0.039 | 0.006 | 0.944 | −0.103 | 0.042 | 0.068 |
| OUTLETAD8 | 0.021 | 0.070 | −0.044 | 0.007 | −0.080 | 0.296 | −0.013 | −0.067 |
| MAXELV | 0.351 | 0.034 | 0.027 | 0.010 | 0.862 | −0.168 | 0.041 | 0.154 |
| RVLEN | −0.120 | 0.176 | 0.623 | −0.270 | 0.061 | −0.067 | −0.191 | −0.008 |
| RVSLP | 0.313 | 0.012 | −0.020 | 0.017 | 0.364 | −0.112 | −0.001 | 0.867 |
| MFP | 0.035 | −0.102 | −0.017 | 0.984 | 0.041 | 0.027 | 0.111 | 0.015 |
| NSTEPS | −0.044 | −0.041 | 0.625 | 0.311 | −0.121 | 0.017 | −0.083 | −0.025 |
| AVEROU | −0.238 | −0.076 | −0.018 | −0.046 | −0.353 | 0.884 | 0.040 | 0.002 |
| AVEINF | −0.019 | 0.006 | −0.063 | −0.042 | −0.058 | −0.097 | 0.013 | −0.031 |

It can be seen from Table 6 that for the first principal component, the 6 h 10-return-year precipitation has the strongest explanatory power; for the second principal component, the 6 h variation coefficient has the strongest explanatory power; for the third principal component, the 1 h 100-return-year modulus coefficient has the strongest explanatory power; and for the fourth principal component, the 10 min 100-return-year precipitation has the strongest explanatory power. The concentration time of small watersheds in hilly areas is usually within 6 h, so 6 h related rainfall factors were selected to represent the rainfall factors of flash floods, specifically 6 h 10-return-year precipitation, 6 h variation coefficient, 1 h 100-return-year modulus coefficient, and 10 min 100-return-year precipitation.

From the component matrix analysis in Table 7, it can be seen that for the first principal component, slope and gradient factors have the strongest explanatory power; for the second principal component, watershed shape coefficient factor has the strongest explanatory power; for the third principal component, concentration time has the strongest explanatory power; and for the fourth principal component, peak modulus has the strongest explanatory power. The confluence of small watersheds in hilly areas is greatly affected by landform, so the average slope, shape coefficient, confluence time and flood peak modulus of the watershed were finally selected as the final terrain factors for risk analysis.

Table 8 shows that for the first principal component, family wealth, has the strongest explanatory power; for the second principal component, reservoir has the strongest explanatory power; and for the third principal component, the number of schools has the strongest explanatory power. The safety of people's lives and property is the primary goal of flood prevention and control, so family wealth, the number of reservoirs, and the number of schools were finally selected as elements at risk.

**Table 8.** Results of Analysis of Elements at Risk.

| Elements at Risk | Com 1 | Com 2 | Com 3 | Elements at Risk | Com 1 | Com 2 | Com 3 |
|---|---|---|---|---|---|---|---|
| PC_A | 0.850 | 0.086 | 0.102 | DAM | 0.027 | 0.400 | 0.030 |
| HTC_A | 0.990 | 0.089 | 0.037 | CULVERT | 0.097 | 0.059 | −0.005 |
| ETC_A | 0.992 | 0.054 | 0.023 | BUSINESS | 0.138 | 0.037 | 0.473 |
| RES | 0.040 | 0.754 | 0.040 | SCHOOL | 0.076 | 0.259 | 0.593 |
| SLUICE | 0.029 | 0.529 | 0.081 | HOSPITAL | 0.037 | 0.000 | 0.333 |
| BRIDGE | 0.012 | 0.269 | 0.087 | NURSINGHOME | −0.005 | 0.001 | 0.083 |

It can be seen from Table 9 that for the first principal component, the weak room proportion factor has the strongest explanatory power; for the second principal component, the flood control capacity has the strongest explanatory power; and for the third principal component, the automatic monitoring site factor has the strongest explanatory power. Considered comprehensively, the vulnerability factors include the proportion of endangered houses, the density of automatic monitoring stations, and the flood control capacity.

**Table 9.** Results of Analysis of Vulnerability Factors.

| Vulnerability | Com 1 | Com 2 | Com 3 | Vulnerability | Com 1 | Com 2 | Com 3 |
|---|---|---|---|---|---|---|---|
| RTC3_A | 0.656 | 0.113 | 0.232 | FHNL | 0.099 | 0.720 | 0.005 |
| RTC4_A | 0.671 | 0.110 | 0.104 | DIKE | 0.034 | 0.042 | 0.097 |
| ZD_WATA | 0.005 | 0.005 | 0.712 | PC_A5 | 0.032 | 0.303 | 0.045 |
| JY_YL | 0.125 | 0.048 | 0.364 | PC_A20 | 0.030 | 0.520 | 0.068 |
| JY_SW | 0.062 | 0.039 | 0.026 | PC_A100 | 0.136 | 0.213 | 0.077 |

### 3.3. Factor Weight Determination

For each factor of 12,321 small watersheds, the natural discontinuity method was used to delineate the parameter interval, and formula 6 was used to calculate the CF value of each factor in each small watershed. Since a total of 11,925 flash floods were recorded this time, and the total area of 12,321 small watersheds was about 215,660 km$^2$, the unified value of *PPs* was set as 0.055. Taking 6 h return-year rainfall as an example, Table 10 shows the data interval and the number of flash floods, total area, *PPa* and *CF* values corresponding to different intervals.

**Table 10.** CF Values of 6 Hour Return-Year Rainfall.

| Item | Range of 6 h Return-Year Rainfall (mm) | | | | |
|---|---|---|---|---|---|
| | ≤110 | 110~120 | 120–130 | 130–140 | >140 |
| Count | 2538 | 747 | 2068 | 3405 | 3169 |
| Area | 43,107 | 22,722 | 47,546 | 46,476 | 55,809 |
| FFa | 0.059 | 0.033 | 0.045 | 0.073 | 0.057 |
| CF | 0.064 | −0.419 | −0.223 | 0.260 | 0.027 |

For each watershed, we could find the CF value according to Table 10 and the 6 h 10-year rainfall of the watershed. Similarly, we could obtain the CF for 15 factors of each watershed. The CF values can be combined by using formula 7. Table 11 shows the combined results for one of the watersheds.

**Table 11.** Z Value Results for One Watershed.

| Item | Value | CF | Item | Z Value |
|---|---|---|---|---|
| $H_{6h\_10\%}$ | 138.64 | 0.260 | $Z_{all\text{-}H6h\_10\%}$ | 0.931 |
| $Cv_{6h}$ | 0.56 | 0.251 | $Z_{all\text{-}Cv6h}$ | 0.932 |
| $KP_{1h\_1\%}$ | 2.96 | 0.136 | $Z_{all\text{-}KP1h\_1\%}$ | 0.941 |
| $H_{10min\_1\%}$ | 38.34 | −0.006 | $Z_{all\text{-}H10min\_1\%}$ | 0.949 |
| WSSLP | 0.13 | 0.222 | $Z_{all\text{-}WSSLP}$ | 0.935 |
| WSSHPC | 0.24 | 0.070 | $Z_{all\text{-}WSSHPC}$ | 0.945 |
| NSTEPS | 3 | −0.014 | $Z_{all\text{-}NSTEPS}$ | 0.950 |
| MFP | 0.107 | 0.004 | $Z_{all\text{-}MFP}$ | 0.949 |
| ETC_A | 1980 | 0.602 | $Z_{all\text{-}ETC\_A}$ | 0.872 |
| RES | 1 | 0.193 | $Z_{all\text{-}RES}$ | 0.937 |
| SCHOOL | 0 | −0.391 | $Z_{all\text{-}SCHOOL}$ | 0.969 |
| RTC_4A | 76.85 | 0.413 | $Z_{all\text{-}RTC\_4A}$ | 0.913 |
| ZD_WATA | 0 | −0.250 | $Z_{all\text{-}ZD\_WATA}$ | 0.962 |
| FHNL | 26 | 0.650 | $Z_{all\text{-}FHNL}$ | 0.855 |

The CF-based multi-factor superposition method was used to determine the weights of factors for flash flood hazards. According to Table 12, we could obtain the distribution of Z-values after removing a certain factor.

**Table 12.** Statistical Percentage of Z-values without Considering a Single Factor.

| The Range of Z-Value | $\leq -0.6$ | $-0.6 \sim -0.2$ | $-0.2 \sim 0.2$ | $0.2 \sim 0.6$ | $> 0.6$ |
|---|---|---|---|---|---|
| $Z_{\text{all- H6h\_10\%}}$ | 0.0 | 27.4 | 49.5 | 23.1 | 0.0 |
| $Z_{\text{all- Cv6h}}$ | 0.0 | 0.0 | 83.8 | 16.2 | 0.0 |
| $Z_{\text{all- KP1h\_1\%}}$ | 0.0 | 42.8 | 39.5 | 17.6 | 0.0 |
| $Z_{\text{all- H10min\_1\%}}$ | 0.0 | 19.3 | 77.2 | 3.5 | 0.0 |
| $Z_{\text{all- WSSLP}}$ | 20.5 | 0.0 | 25.1 | 54.4 | 0.0 |
| $Z_{\text{all- WSSHPC}}$ | 0.0 | 0.0 | 100.0 | 0.0 | 0.0 |
| $Z_{\text{all-NSTEPS}}$ | 0.0 | 0.0 | 100.0 | 0.0 | 0.0 |
| $Z_{\text{all-MFP}}$ | 0.0 | 0.0 | 97.3 | 0.0 | 0.0 |
| $Z_{\text{all-ETC\_A}}$ | 52.4 | 0.0 | 10.9 | 13.0 | 23.5 |
| $Z_{\text{all-RES}}$ | 0.0 | 0.0 | 100.0 | 0.0 | 0.0 |
| $Z_{\text{all- SCHOOL}}$ | 0.0 | 81.6 | 0.0 | 0.0 | 18.4 |
| $Z_{\text{all- RTC\_4A}}$ | 53.9 | 0.0 | 0.0 | 31.3 | 14.7 |
| $Z_{\text{all- ZD\_WATA}}$ | 0.0 | 83.3 | 0.0 | 12.7 | 4.0 |
| $Z_{\text{all-FHNL}}$ | 0.0 | 86.4 | 0.0 | 1.9 | 11.6 |

Considering all factors, we could compute the distribution of $Z_{\text{all}}$. The contribution of each factor was calculated and normalized to obtain its weight. Table 13 shows the weight of each risk factor.

**Table 13.** Weight of Each Risk Factor.

| | Risk Factors | | Sum of Absolute Values | Weight |
|---|---|---|---|---|
| Hazard Factors (0.66) | Rainfall Factors (0.26) | $H_{6h\_10\%}$ | 2.1 | 0.05 |
| | | $Cv_{6h}$ | 4.5 | 0.12 |
| | | $KP_{1h\_1\%}$ | 2.3 | 0.06 |
| | | $H_{10min\_1\%}$ | 1.2 | 0.03 |
| | Terrain Factors (0.40) | WSSLP | 3.1 | 0.08 |
| | | WSSHPC | 4.4 | 0.11 |
| | | NSTEPS | 4.2 | 0.11 |
| | | MFP | 4.0 | 0.10 |
| Elements at Risk (0.20) | ETC_A | | 1.1 | 0.03 |
| | RES | | 4.0 | 0.10 |
| | SCHOOL | | 2.8 | 0.07 |
| Vulnerability Factors (0.14) | RTC_4A | | 1.7 | 0.04 |
| | ZD_WATA | | 1.7 | 0.05 |
| | FHNL | | 1.8 | 0.05 |

*3.4. Results*

As shown above, 14 factors were selected to form the flash flood risk indicator system. Using the identification technology of flash flood disaster risk factors, the risk index of small watersheds in Hubei Province was calculated according to the weight index determined by the risk assessment model and considering the impacts of hazards, elements at risk and vulnerability, respectively. The risk index was classified according to the natural discontinuity method to obtain the risk level of each small watershed. Figure 9 shows the risk distribution of flash floods in Hubei Province.

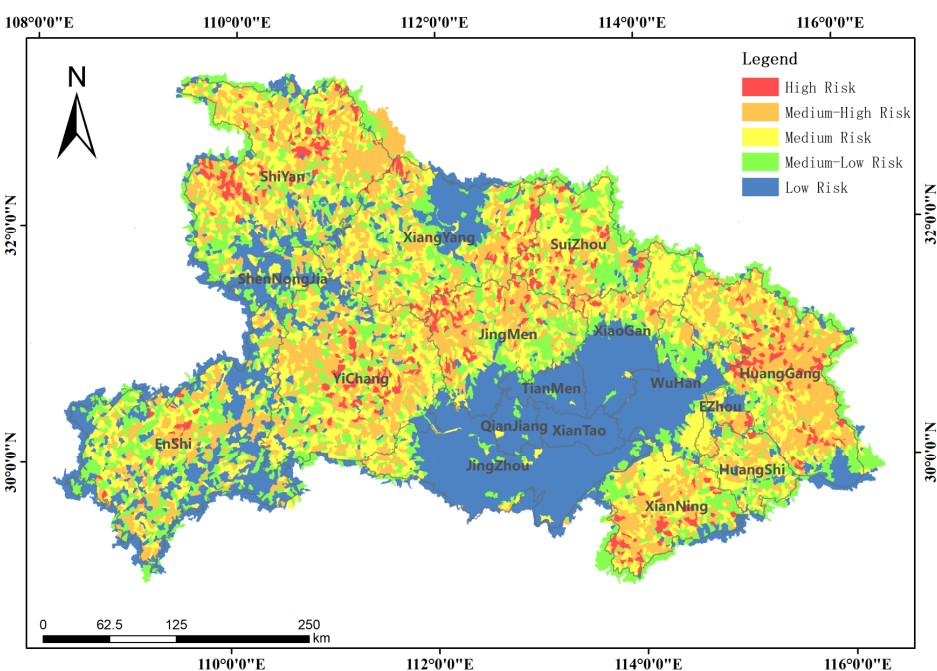

**Figure 9.** Risk Distribution of Flash Floods in Hubei Province.

It can be seen from Figure 9 and Table 14 that the medium-high risk areas in Huang-gang accounted for 47.00%, and high-risk areas accounted for 8.70%, with both areas adding up to more than 50%, followed by more than 40% in Shiyan, E'zhou and Xianning and more than 30% in Huangshi, Yichang, Xiangyang, Jingmen and Suizhou. Comparing with Figure 4, it can be seen that the high-risk areas correspond well to those with high flash flood frequencies.

**Table 14.** The Proportion of Risk Areas of Different Levels in Each City.

| City | Risk Level | | | | |
|------|-----|------------|--------|-------------|------|
| | **Low** | **Medium-Low** | **Medium** | **Medium-High** | **High** |
| Wuhan | 55.40% | 18.90% | 17.50% | 7.20% | 0.90% |
| Huangshi | 8.70% | 26.30% | 34.50% | 29.30% | 1.10% |
| Shiyan | 10.50% | 19.20% | 28.60% | 34.70% | 7.00% |
| Yichang | 9.80% | 19.30% | 31.40% | 34.00% | 5.50% |
| Xiangyang | 20.90% | 20.40% | 27.70% | 27.00% | 4.00% |
| E'zhou | 11.20% | 8.60% | 39.60% | 34.50% | 6.00% |
| Jingmen | 13.70% | 22.20% | 26.00% | 29.90% | 8.20% |
| Xiaogan | 49.80% | 13.10% | 25.80% | 9.70% | 1.60% |
| Jingzhou | 83.60% | 7.00% | 6.50% | 3.00% | 0.00% |
| Huanggang | 7.50% | 12.00% | 24.90% | 47.00% | 8.70% |
| Xianning | 6.90% | 19.00% | 32.10% | 33.10% | 8.80% |
| Suizhou | 2.30% | 31.90% | 35.00% | 22.10% | 8.80% |
| Enshi | 31.60% | 24.80% | 21.20% | 21.00% | 1.40% |
| Xiantao | 100.0% | 0.00% | 0.00% | 0.00% | 0.00% |
| Qianjiang | 92.50% | 5.10% | 2.40% | 0.00% | 0.00% |
| Tianmen | 93.80% | 5.30% | 0.90% | 0.00% | 0.00% |
| Shennongjia | 49.80% | 21.90% | 16.50% | 11.80% | 0.00% |

See Table 15 for statistics on the distribution of risk levels in the 11 hydro-meteorological regions. The medium-high and high risk areas together account for 73.82% of District I, exceed 40% of Districts II, VIII and IX, and exceed 30% of District VI.

**Table 15.** The Proportion of Risk Areas of Different Levels in Hydro-meteorological Regions.

| Hydro-Meteorological Regions | Risk Level | | | | |
|---|---|---|---|---|---|
| | Low | Medium-Low | Medium | Medium-High | High |
| I | 0.69% | 6.82% | 18.67% | 60.94% | 12.88% |
| II | 4.17% | 21.19% | 29.88% | 33.77% | 10.99% |
| III | 46.90% | 14.91% | 20.40% | 15.71% | 2.09% |
| IV | 13.16% | 25.28% | 32.36% | 24.89% | 4.31% |
| V | 85.38% | 7.41% | 5.44% | 1.59% | 0.19% |
| VI | 17.24% | 19.94% | 27.64% | 27.41% | 7.77% |
| VII | 19.85% | 21.44% | 33.32% | 22.73% | 2.66% |
| VIII | 9.89% | 17.78% | 28.41% | 37.38% | 6.53% |
| IX | 4.97% | 17.55% | 33.48% | 37.64% | 6.36% |
| X | 28.60% | 25.27% | 22.24% | 22.43% | 1.46% |
| XI | 24.46% | 22.77% | 23.29% | 24.61% | 4.86% |

## 4. Discussions

### 4.1. Distribution of Flash Flood Risks

It can be seen from Figure 9 that the spatial distribution of flood disaster risks in mountainous areas of Hubei Province manifests significant characteristics: the flash flood risk in the eastern and western regions of Hubei Province is generally higher than that in the central region [28], mainly because of the predominantly mountainous and hilly terrain in the eastern and western regions. In contrast, the central region, especially the south-central region of Hubei Province, is dominated by flat low mountains and hills, with little variation in elevation. The risk of flash floods in the central region is relatively low, and even heavy rain is unlikely to cause flooding in a short period of time, thus unlikely to meet the prerequisites for forming flash flood disasters.

As shown in Figure 4, Shiyan, Yichang, Enshi, Xiangyang, Huanggang, and Xianning are all flash flood-prone areas. The density and frequency of flash floods in some areas are relatively high, which is consistent with the risk analysis results. It should be noted that the risk value calculated in Ezhou is relatively high, but the impact of flash floods here is actually very small. The analysis results for Ezhou are not completely consistent with the actual situation, which is an important topic that requires further investigation. It can be seen from Figure 1 that Ezhou is affected by a wide range of disasters, but the disasters are not serious and their destructive power is relatively weak. This may be due to the fact that the number of flash flood disasters is included in the calculation without distinguishing the impact of flash flood disasters.

### 4.2. Discussion of Driving Factor Results

It can be seen from Table 13 that the total weight of hazard factors was 0.66, including 0.26 for the precipitation factor and 0.40 for the terrain factor, while the weights of the elements at risk and the vulnerability factor were 0.20 and 0.14, respectively. Among the 14 risk factors, the 6 h coefficient of variation had the highest weight, of 0.12, followed by the watershed shape coefficient and concentration time, with a weight of 0.11; the watershed flood peak modulus and reservoir quantity, with a weight of 0.10; the sum of the weights of these factors exceeded 0.5. These factors can be considered as the main causes of flash flood disasters in Hubei Province.

Possible reasons for the variations in weights were mainly because heavy rain was the direct premise of causing flash floods, and the 6 h coefficient of variation had the highest weight; the concentration time and flood peak modulus were direct parameters that reflected the concentration, while the shape coefficient was an indirect parameter, as fan-shaped basins are more likely to generate large floods than narrow or long basins under the same conditions.

Furthermore, such a high weight for reservoir quantity was somewhat difficult to explain. The reservoirs in Hubei Province are mainly small with a storage capacity of less

than 10 million m$^3$, scattered all over the province. The distribution of small reservoirs is somewhat similar to the layout of residential settlements, as the construction of hydraulic engineering facilities is determined by the needs of human activities. In addition, before the large-scale reinforcement in Hubei Province works, low construction standards and poor management made some small reservoirs more likely to cause flood disasters.

### 4.3. Prospects and Limitations

With the support of the Flash Flood Disasters Investigation and Evaluation Dataset, 98 risk factors were considered. We compared the dataset with other international studies [29–31], but such a rich dataset is rare. However, in China, this dataset is available in every province, so the methodology of this study can be easily published in China.

Outside of China, this method has data limitations and is not suitable for direct application. The Pearson correlation coefficient method and principal component analysis method can be used for factor analysis and dimension reduction processing in any field. The CF-based multi-factor superposition method adopted in this paper was initially applied to the risk assessment of geological disasters in China. The study area included plateaus, mountains, hills and watersheds, involving a variety of landforms and complex geological structures with many similarities to areas of flash flooding in China. The CF-based multi-factor superposition method can be widely used in the risk assessment of flash flood disasters in China, with great reference value and significance. In addition, the CF-based multi-factor superposition method has higher requirements for the dataset, especially the historical flood investigation data, as it enables one to roughly understand the distribution of disaster frequency, which is very important for determining *PPs*, defined in Section 2.3.2.

### 5. Conclusions

In this study, a risk model of flash flood disasters was established according to the theory of natural disaster systems. By screening out the risk factors related to flash floods and analyzing the weight of each factor, the risk distribution of flash floods in Hubei Province was determined. The main results are as follows:

(1) Based on the Pearson correlation coefficient method and principal component analysis method, 14 risk factors were selected from 98 factors to establish a risk assessment model. The weight of each risk factor was determined by the CF-based multi-factor superposition method. Among the 14 risk factors, the 6 h variation coefficient had the highest weight, of 0.12, followed by the watershed shape coefficient and concentration time, both 0.11, as well as flood peak modulus and number of reservoirs, both 0.1, and the total weight of these five factors exceeded 0.5.

(2) The risk distribution was highly consistent with the location of flash floods. The medium-high risk areas in Huanggang accounted for 47.00%, and high-risk areas accounted for 8.70%, with both areas adding up to more than 50%, followed by more than 40% in Shiyan, Ezhou and Xianning, and more than 30% in Huangshi, Yichang, Xiangyang, Jingmen and Suizhou.

(3) The trends in the distribution of risk levels in this study were generally consistent with the historical flood distribution of the investigation results, indicating that the investigation results had a high degree of confidence. Hubei Province has a lack of research results on flash flood disaster risks, so we only briefly described the risk distribution, which was quite consistent with the investigation results, but it was not further analyzed in this paper. How to quantify and scientifically demonstrate the rationality of risk results is another issue that we need to discuss. Due to changes in natural conditions caused by social and economic development and other human activities, the key factors contributing to the risks of flash flood disasters have been constantly changing. Therefore, it is necessary to continue disaster investigations in flood-prone areas and conduct proper analyses to ensure the accuracy of the risk results.

(4)    The dataset used in this study was mainly based on investigation and evaluation results from 2013 to 2015. Since 2015, Hubei Province has carried out some investigation and evaluation work to supplement and improve the dataset. However, due to the incomplete integration of these data, this part of the dataset was temporarily ignored in this study. Follow-up studies should further refine the baseline dataset to improve the accuracy of the risk results.

(5)    Subsequent calculations of soil water content should be combined with numerical precipitation forecasts, short-term radar precipitation forecasts and actual precipitation data; the likelihood of flash floods should be assessed according to the risk levels of flash floods in different areas to provide a supporting basis for flood prevention and mitigation. In addition, further research should be carried out on the application of risk results, including the interface with the dynamic management inventory for flood hazard areas, the appropriate allocation of project funds, etc.

**Author Contributions:** Conceptualization, Y.T., H.W.; methodology, Y.T., Q.M.; software, Y.Z.; validation, R.D., Y.Z.; project administration, B.H., C.L. All authors have read and agreed to the published version of the manuscript.

**Funding:** This research was funded by National Key Research Program (2019YFC1510603), Hubei Flash Flood Prevention Project (SLT-SHZH-2022).

**Institutional Review Board Statement:** Not applicable.

**Informed Consent Statement:** Not applicable.

**Conflicts of Interest:** The authors declare no conflict of interest.

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
