# Peer review of "Study on Risk Assessment of Flash Floods in Hubei Province"

_water, doi:10.3390/w15040617_

Round 1
Reviewer 1 Report
Review of manuscript "Study on Risk Assessment of Flash Floods in Hubei Province" (water-2169894)
Dear authors, your research aims to establish the risk model of flash flood disaster according to the theory of natural disaster system in Hubei Province, China. It is a well-prepared manuscript and fits the aims and scope of the journal topic. Nevertheless, the authors need to highlight the soundness and novelty of their research as compared with previous research. Therefore, "Major Revision" is necessary to improve this manuscript. Specifically, the reviewer has the following comments and suggestions:
(1) The Abstract and Introduction Section: these two parts are not strong because the authors did not highlight the necessity and novelty of this study from an international perspective. As a consequence, reviewers cannot figure out why this research must be performed in this context. If this research just presented a case study in a particular region, namely, the Hubei Province, then it lacks enough novelty for publishing in this internationally distinguished journal (Water). I would like to remind that these several methods, namely, the Risk Conceptual Model, Pearson correlation coefficient and principal component analysis method are not new in flood risk assessment.
(2) At the end of the Introduction, the authors have mentioned a number of studies related to flood risk assessment, but without mentioning the disadvantages of these studies.
(3) Figure 1: please explain the meaning of those villages. I think the flash flood hazard points should be instead presented in this figure.
(4) The basic data set and result data set of flood investigation and evaluation in Hubei Province from 2013 to 2015 were collected. Why only 2013 to 2015 were considered?
(5) Section 2.2. Data Collection: in this part, the authors did not present all the details of the input data, such as the dates in acquiring them, accuracies, temporal and spatial resolutions. The authors should largely improve their statements and expressions.
(6) What about the non-flood samples? A random sampling strategy may result in great uncertainties of the flooding susceptibility modelling. The lack of accurate absence data will inevitably increase the difficulty of flood risk assessment.
(7) Section 3.1. Risk Factor Identification: Please provide references to support why these factors have been taken into account in this study.
(8) Did the authors consider and deal with the problem of multi-collinearity?
(9) The Literature Part: in this part, the authors need to look further into the relevant research about future flood risk prediction. In particular, the advanced maximum entropy method has been largely used in flooding susceptibility assessment (please find below). Nevertheless, this new method was ignored in this manuscript. A thorough literature review is meant to set the context for your research work and highlight how it contributes to the knowledge in this field and builds on previous relevant research.
https://doi.org/10.1016/j.scs.2022.103812
https://doi.org/10.1080/10106049.2017.1316780
https://doi.org/10.1007/s11069-020-04453-3
(10) Are all those risk factors obtained from 2013 to 2015 because the historical large flood events occurred from 2013 to 2015?
(11) Table 1. Division of CF value: how to determine the range of CF value?
(12) Please also provide the detailed processes for the determination of the key parameters for running those mathematical methods.
(13) The results, in particular, failed to highlight the value of using this approach compared to traditional assessments. There are few quantified comparisons that show differences or improvements over the more common models. I think the authors need to reinforce what are the roles of the Pearson correlation coefficient and principal component analysis method.
(14) The authors also need to improve the Conclusion part by clarifying the main shortages of your work.
Author Response
Reply to reviewers’ comments on WATER-2169894
We would like to express our sincere thanks to the editor and reviewers for their valuable evaluation and useful suggestions on our paper,which are valuable in improving the quality of our manuscript. We have made careful modifications on our manuscript in the related parts (refer to the revised manuscript) in accordance with the comments and suggestions of Reviewer #1 and Reviewer #2. In order to make the revisions more understandable, we provide a revised version of the document. The followings are the summary of how we revised the manuscript in response to the reviewers’ comments.
Response to Reviewer 1 Comments
Point 1: The Abstract and Introduction Section: these two parts are not strong because the authors did not highlight the necessity and novelty of this study from an international perspective. As a consequence, reviewers cannot figure out why this research must be performed in this context. If this research just presented a case study in a particular region, namely, the Hubei Province, then it lacks enough novelty for publishing in this internationally distinguished journal (Water). I would like to remind that these several methods, namely, the Risk Conceptual Model, Pearson correlation coefficient and principal component analysis method are not new in flood risk assessment.
Response 1: The Flash Flood Disasters Investigation and Evaluation project has been the largest non-engineering water conservancy projects in China since the founding of new China. It is also the largest-scale general census on disaster background in flood management and mitigation fields. The whole project lasted from 2012 to 2016 for 4 years, covering 30 provinces, 305 cities and 2138 counties, involving a total land area of 7.55 million km2 and a population of nearly 900 million. By investing a lot of human, material and financial resources and using various means such as survey, measurement and analysis, the project collected the scope of China's flash flood disaster prevention and control areas, the distribution of people, underlying surface conditions, socio-economics, historical flash floods and other basic information, scientifically analyzed the characteristics of flash floods in small watersheds in hilly areas, evaluated the current flood control capacity, calculated early warning indicators, delineated the flood hazard areas, and on this basis established a scientific data set for flash flood disaster investigation and evaluation, which provided basic information support for flash flood disaster early warning and forecasting and emergency rescue decisions.
Extensive flood investigation and risk assessment were carried out in Hubei Province, and the investigation and evaluation work was mainly carried out and completed in 2013-2015. In the early stage, the villages that have suffered from flood disasters since 1949 and have not suffered from flood disasters but are close to rivers were selected from the whole province. Site surveys and measurements were carried out in these villages, and the flood risks of these villages were assessed. These analysis samples can cover most of the areas threatened by flash floods in Hubei Province, but affected by technical and non-technical factors, these data cannot comprehensively reflect the flash flood risks in the province. Some additional work was carried out after 2015, but the results have not yet been systematically aggregated .
On the basis of the conceptual risk model, combining the advantages of Pearson correlation coefficient, principal component analysis, factor analysis, CF-based multi-factor superposition method, etc., the risks of Hubei Province were quantified and classified, the factors with the greatest impact on flash floods in Hubei Province were extracted and their weights were analyzed, and the risk levels of different regions were scientifically and quantitatively determined, with significant reference value for flood risk analysis in other regions of China.
Point 2: At the end of the Introduction, the authors have mentioned a number of studies related to flood risk assessment, but without mentioning the disadvantages of these studies.
Response 2: We added a description at the end of the introduction. The selection of elements for risk assessment is often subjective and limited by the basic data set. This study aims to create a set of scientific, quantitative and replicable risk assessment processes and standards for flash flood risk assessment in China, based on the data set of Flash Flood Disasters Investigation and Evaluation results. Therefore, it is inappropriate to directly reuse the findings of other studies, which lack relevance and applicability.
Point 3: Figure 1: please explain the meaning of those villages. I think the flash flood hazard points should be instead presented in this figure.
Response 3: We have updated the illustrations. These villages are the key targets for the prevention of flash floods, and where the investigation and evaluation work was carried out. They are located in the hilly area and close to the river channels, so they are more likely to suffer from flash floods. Some of the villages suffered one or more flash floods during 1949-2015.
Point 4: The basic data set and result data set of flood investigation and evaluation in Hubei Province from 2013 to 2015 were collected. Why only 2013 to 2015 were considered?
Response 4: I'm sorry that the explanation here is not clear, which has troubled your reading. The investigation in Hubei Province was carried out from 2013 to 2015. In fact, the flood samples was taken from 1949 to 2015. Subsequently, some investigation and evaluation work has been carried out since 2015 to fill gaps in previous projects, but the results have not yet been systematically aggregated.
Point 5: Section 2.2. Data Collection: in this part, the authors did not present all the details of the input data, such as the dates in acquiring them, accuracies, temporal and spatial resolutions. The authors should largely improve their statements and expressions.
Response 5: I'm sorry that the explanation here is too simple. The investigation work in Hubei Province has been additionally described in the previous section and will not be repeated here. And we have rewritten Section 2.2. Thank you for your valuable comments.
In 2013-2015, a total of 74 counties (cities and districts) in Hubei Province were investigated and evaluated, involving a total population of 42.114 million and an area of 160,000 km2. The total population in the control area was 16.34 million, and the total land area was 126,700 km2. The investigation covered 44,000 flash flood hazard areas in the province, with a total population of 3.92 million, involving 7,189 enterprises and institutions, and 1.12 million households in hazard areas. On this basis, social and economic surveys were conducted on 44,608 villages in the control area, and the regional distribution of 1,544 historical floods, 3,240 automatic monitoring stations, 8,868 wireless early warning transmitting stations, 794 simple hydrological gauging stations, and 12,012 simple rainfall gauging stations were analyzed. Moreover, detailed surveys and investigations were carried out on the riverside villages, and 11,818 longitudinal sections and 35,318 cross sections of ditches where riverine villages are located were measured and collated, providing valuable basic data for the prevention and control of flash flood disasters in Hubei Province.
Point 6: What about the non-flood samples? A random sampling strategy may result in great uncertainties of the flooding susceptibility modelling. The lack of accurate absence data will inevitably increase the difficulty of flood risk assessment.
Response 6: In the initial stage of the flash flood disasters investigation in Hubei Province, a systematic inventory of villages that have experienced flash floods and villages that have not experienced flash floods but may be affected by them was conducted. Therefore, this is a relatively comprehensive and meticulous work, and the samples collected are very rich, including flood samples and non-flood samples, not just sampling.
Point 7: Section 3.1. Risk Factor Identification: Please provide references to support why these factors have been taken into account in this study.
Response 7: Thank you very much for your suggestion. We have added relevant references here.
Point 8: Did the authors consider and deal with the problem of multi-collinearity?
Response 8: The Pearson correlation coefficient method was mainly used to eliminate the influence of multi-collinearity. In the final factor screening results, the factors with weak linear correlation were retained.
Point 9: The Literature Part: in this part, the authors need to look further into the relevant research about future flood risk prediction. In particular, the advanced maximum entropy method has been largely used in flooding susceptibility assessment (please find below). Nevertheless, this new method was ignored in this manuscript. A thorough literature review is meant to set the context for your research work and highlight how it contributes to the knowledge in this field and builds on previous relevant research.
https://doi.org/10.1016/j.scs.2022.103812
https://doi.org/10.1080/10106049.2017.1316780
https://doi.org/10.1007/s11069-020-04453-3
Response 9: Thank you very much for your valuable comments, we have added relevant references here. The maximum entropy method is often used for relatively simple environmental risk assessment of the underlying surface (such as urban waterlogging, drought assessment, etc.), which is very different from the initial environment of flash flood disasters.
Point 10: Are all those risk factors obtained from 2013 to 2015 because the historical large flood events occurred from 2013 to 2015?
Response 10: I'm sorry again that the explanation here is not clear, which has troubled your reading. The investigation in Hubei Province was carried out from 2013 to 2015. In fact, the flood samples were taken from 1949 to 2015. We have rewritten Section 2.2. Thank you for your valuable comments.
Point 11: Table 1. Division of CF value: how to determine the range of CF value?
Response 11: The CF value is determined by formula 6, and we have added specific instructions in the manuscript.
The certainty factor model (CF) was used to analyze the weight of each factor. This model was originally developed by Shortliffe et al.[23], and improved by Heckerman [24]. The formula is as follows:
(6)
Where, PPa is the conditional probability of occurrence of events (flash flood disasters) in Type-a data, and is the ratio of the number of flash flood disasters in Type-a data to the area of Type-a; PPs is the prior probability of flash flood disasters in the entire study area, which can be expressed as the ratio of the number of flash flood disasters in the entire study area to the area of the study area.
For each factor of 12,321 small watersheds, the natural discontinuity method was used to delineate the parameter interval, and formula 6 was used to calculate the CF value of each factor in each small watershed. Since a total of 11,925 flash floods were recorded this time, and the total area of 12,321 small watersheds was about 215,660 km2, the unified value of PPs was set as 0.055. Taking 6-hour -return-year rainfall as an example, Table 10 shows the data interval, and the number of flash floods, total area, PPa and CF values corresponding to different intervals.
Table 10. CF Values of 6-Hour -Return-Year Rainfall
Item |
Range of 6-hour -return-year Rainfall(mm) |
||||
≤110 |
110~120 |
120-130 |
130-140 |
>140 |
|
Count |
2,538 |
747 |
2,068 |
3,405 |
3,169 |
Area |
43,107 |
22,722 |
47,546 |
46,476 |
55,809 |
FFa |
0.059 |
0.033 |
0.045 |
0.073 |
0.057 |
CF |
0.064 |
-0.419 |
-0.223 |
0.260 |
0.027 |
Point 12: Please also provide the detailed processes for the determination of the key parameters for running those mathematical methods.
Response 12: Thank you very much for your valuable comments. Some detailed calculation examples are added in the manuscript to help readers understand the process more easily. See section 3.3 of the manuscript for details.
Point 13: The results, in particular, failed to highlight the value of using this approach compared to traditional assessments. There are few quantified comparisons that show differences or improvements over the more common models. I think the authors need to reinforce what are the roles of the Pearson correlation coefficient and principal component analysis method.
Response 13: At present, there are few studies based on the investigation and evaluation result data set in China, and Hubei Province is also lack of research results on flash flood disaster risks. The purpose of this study is to find a feasible risk assessment method based on the Flash Flood Disasters Investigation and Evaluation dataset, reveal the cause of flash flood disaster formation to a certain extent, and identify the key factors affecting flood disasters by making full use of the existing data. Pearson correlation coefficient mainly analyzes the linear correlation between several factors. Principal component analysis is to describe the most important characteristics of the data with the least data. These calculations are to reduce the data dimension and eliminate the collinearity impact of the data. Through a series of dimensionality reduction processing, it is helpful for us to find the formation rules of flash floods from the basic data. The CF-based multi-factor superposition method adopted in this manuscript was initially applied to the risk assessment of geological disasters in China. Its research areas include plateaus, mountains, hills and watersheds, involving a variety of landforms and complex geological structures, which has many similarities with China's flash floods.
Point 14: The authors also need to improve the Conclusion part by clarifying the main shortages of your work.
Response 14: The data set currently used is mainly based on the investigation and evaluation results from 2013 to 2015. Since 2015, Hubei Province has carried out some work to supplement and improve the data set. However, due to the incomplete integration of these data, this part of the data was temporarily ignored in this study. Follow-up studies should further refine the baseline dataset to improve the accuracy of the results.
Hubei Province is lack of research results on flash flood disaster risks, so the manuscript only briefly describes the distribution of risk results, which is quite consistent with our investigation results, but it has not been further analyzed in the manuscript. How to quantify and scientifically demonstrate the rationality of risk achievements is another topic that we need to discuss.
Further research should be carried out on the application of risk results, including the interface with the dynamic management inventory for flood hazard areas, and the appropriate allocation of project funds, etc.
The above responses are our reply to reviewers’ comments on WATER-2169894. We look forward to hearing further information from you.
Sincerely yours,

Reviewer 2 Report
Dear authors, my corrections as following:
1. Lines 28-30:
They affect worldwide, please add references from different regions of the world such as the following ones:
A conceptual flash flood early warning system for Africa, based on terrestrial microwave links and flash flood guidance. ISPRS international journal of geo-information, 3(2), 584-598.
Flood risk-related research trends in Latin America and the Caribbean. Water, 14(1), 10.
Natural hazards in Australia: floods. Climatic Change, 139(1), 21-35.
Natural disaster risk inequalities in Central America. Papers in Applied Geography, 1-15.
Compound flood impact forecasting: integrating fluvial and flash flood impact assessments into a unified system. Hydrology and Earth System Sciences, 26(3), 689-709.
Dendrogeomorphology as a tool to depict hydrogeomorphic processes in the tropics. Revista Cartográfica, (106), 35-51.
2. Lines 79-85:
Here, you have to add the hypothesis of your work, add the aims, and finally indicate the implications of your study in China, your region, and similar countries worldwide
3. Figure 1 and all map figures:
Please add extreme geographic coordinates to all your maps. Moreover, your data (region) must appear in a geographical context, not as an island. Please add surrounding regions with a satellite image behind or something else, but not as an island.
4. Line 125 and ahead:
Your model is based in previous works or you are proposing a complete brand new model??
You need to add some references of previous studies where you based your method.
5. Line 165:
In this subsection is the same, please add references of previous works you based your methods.
6. Line 404:
A Discussion section is needed:
a. Discuss your results, compare with regional/local studies and go further with similar studies in other regions of the world
b. Which are the global implications of your results?
In this section you must cite different studies dealing with flood risk assessments. Here, I highly recommend the following ones:
Advances and challenges in flash flood risk assessment: A review. Journal of Geography & Natural Disasters, 7(2), 1-6.
Flood risk index development at the municipal level in Costa Rica: A methodological framework. Environmental Science & Policy, 133, 98-106.
Methodology of flood risk assessment from flash floods based on hazard and vulnerability of the river basin. Natural Hazards, 79(3), 2055-2071.
Flash flood impacts of Hurricane Otto and hydrometeorological risk mapping in Costa Rica. Geografisk Tidsskrift-Danish Journal of Geography, 120(2), 142-155.
All the best.
Author Response
Reply to reviewers’ comments on WATER-2169894
We would like to express our sincere thanks to the editor and reviewers for their valuable evaluation and useful suggestions on our paper,which are valuable in improving the quality of our manuscript. We have made careful modifications on our manuscript in the related parts (refer to the revised manuscript) in accordance with the comments and suggestions of Reviewer #1 and Reviewer #2. In order to make the revisions more understandable, we provide a revised version of the document. The followings are the summary of how we revised the manuscript in response to the reviewers’ comments.
Response to Reviewer 2 Comments
Point 1: Lines 28-30:
They affect worldwide, please add references from different regions of the world such as the following ones:
A conceptual flash flood early warning system for Africa, based on terrestrial microwave links and flash flood guidance. ISPRS international journal of geo-information, 3(2), 584-598.
Flood risk-related research trends in Latin America and the Caribbean. Water, 14(1), 10.
Natural hazards in Australia: floods. Climatic Change, 139(1), 21-35.
Natural disaster risk inequalities in Central America. Papers in Applied Geography, 1-15.
Compound flood impact forecasting: integrating fluvial and flash flood impact assessments into a unified system. Hydrology and Earth System Sciences, 26(3), 689-709.
Dendrogeomorphology as a tool to depict hydrogeomorphic processes in the tropics. Revista Cartográfica, (106), 35-51.
Response 1: Thank you for your valuable comments. We refer to the above documents in the introduction to supplement the corresponding contents.
Point 2: Lines 79-85:
Here, you have to add the hypothesis of your work, add the aims, and finally indicate the implications of your study in China, your region, and similar countries worldwide
Response 2: Thank you very much for your valuable comments. The purpose of this study is to find a feasible risk assessment method based on the Flash Flood Disasters Investigation and Evaluation dataset, reveal the cause of flash flood disaster formation to a certain extent, and identify the key factors affecting flood disasters by making full use of the existing data. Finally, a set of scientific, quantitative and replicable risk assessment processes and standards for flash flood risk assessment in China will be tailored, based on the data set of Flash Flood Disasters Investigation and Evaluation results.
Point 3: Figure 1 and all map figures:
Please add extreme geographic coordinates to all your maps. Moreover, your data (region) must appear in a geographical context, not as an island. Please add surrounding regions with a satellite image behind or something else, but not as an island.
Response 3: Thank you very much for your valuable comments. According to the suggestions, we have supplemented the Figure containing the location of Hubei Province, and added coordinates for figures in the manuscript, including Figure 1-5 and Figure 8.
Point 4: Line 125 and ahead:
Your model is based in previous works or you are proposing a complete brand new model??
You need to add some references of previous studies where you based your method.
Response 4: Thank you for your valuable comments. In this part, we refer to the theoretical methods of others, and we supplement the references in the corresponding part of the manuscript.
Point 5: Line 165:
In this subsection is the same, please add references of previous works you based your methods.
Response 5: Thank you very much for your valuable comments. In this part, we refer to the theoretical methods of others, and we supplement the references in the corresponding part of the manuscript.
Point 6: Line 404:
A Discussion section is needed:
a. Discuss your results, compare with regional/local studies and go further with similar studies in other regions of the world
b. Which are the global implications of your results?
In this section you must cite different studies dealing with flood risk assessments. Here, I highly recommend the following ones:
Advances and challenges in flash flood risk assessment: A review. Journal of Geography & Natural Disasters, 7(2), 1-6.
Flood risk index development at the municipal level in Costa Rica: A methodological framework. Environmental Science & Policy, 133, 98-106.
Methodology of flood risk assessment from flash floods based on hazard and vulnerability of the river basin. Natural Hazards, 79(3), 2055-2071.
Flash flood impacts of Hurricane Otto and hydrometeorological risk mapping in Costa Rica. Geografisk Tidsskrift-Danish Journal of Geography, 120(2), 142-155.
Response 6: Thank you very much for this comment. At present, there are few studies based on the data set of investigation and evaluation results in China, and there is also a lack of research results on flash flood disaster risks in Hubei Province. The Flash Flood Disasters Investigation and Evaluation project has been the largest non-engineering water conservancy projects in China since the founding of new China. It is also the largest-scale general census on disaster background in flood management and mitigation fields. The whole project lasted from 2012 to 2016 for 4 years, covering 30 provinces, 305 cities and 2138 counties (districts), involving a total land area of 7.55 million km2 and a population of nearly 900 million. By investing a lot of human, material and financial resources and using various means such as survey, measurement and analysis, the project collected the scope of China's flash flood disaster prevention and control areas, the distribution of people, underlying surface conditions, socio-economics, historical flash floods and other basic information, scientifically analyzed the characteristics of flash floods in small watersheds in hilly areas, evaluated the current flood control capacity, calculated early warning indicators, delineated the flood hazard areas, and on this basis established a scientific data set for flash flood disaster investigation and evaluation, which provided basic information support for flash flood disaster early warning and forecasting and emergency rescue decisions.
Hubei Province is lack of research results on flash flood disaster risks, so we only briefly describe the distribution of risk results, which is quite consistent with investigation results, but it has not been further analyzed in this manuscript. How to quantify and scientifically demonstrate the rationality of risk results is another issue that we need to discuss.
The purpose of this study is to find a feasible risk assessment method based on the Flash Flood Disasters Investigation and Evaluation dataset, reveal the cause of flash flood disaster formation to a certain extent, and identify the key factors affecting flood disasters by making full use of the existing data, and provide reference and basis for flash flood risk assessment in other provinces in China. Pearson correlation coefficient mainly analyzes the linear correlation between several factors. Principal component analysis is to describe the most important characteristics of the data with the least data. These calculations are to reduce the data dimension and eliminate the collinearity impact of the data. Through a series of dimensionality reduction processing, it is helpful for us to find the formation rules of flash floods from the basic data. The CF-based multi-factor superposition method adopted in this manuscript was initially applied to the risk assessment of geological disasters in China. Its research areas include plateaus, mountains, hills and watersheds, involving a variety of landforms and complex geological structures, which has many similarities with China's flash floods. The method can be widely used in the risk assessment of flash flood disasters in China, with great reference value and significance.
The above responses are our reply to reviewers’ comments on WATER-2169894. We look forward to hearing further information from you.
Sincerely yours,

Round 2
Reviewer 1 Report
I appreciate the authors' efforts to improve this manuscript. Now it is acceptable for publication.
Author Response
Response to Editor Decision
Point 1 : The term "bearing body" is confusing and should be clarified. Is it the same as "element at risks"? If yes, you may replace it.
Response 1: As you say, the “bearing body” is really confusing. We think the “elements at risk” you put forward is more in line with the usual expressions. We incorporated this suggestion into the manuscript and revised and updated all text, tables and figures.
Point 2: Why no Discussion section is added as proposed by reviewer2?
Response 2: I am sorry for not responding to this suggestion in the first time. We have added Discussion section including distribution of the flash flood risk, discussion of driving factors results, prospects and limitations, as described in the revised manuscript.
The above responses are our reply to reviewers’ comments on WATER-2169894. We look forward to hearing further information from you.
Sincerely yours,
All authors.
